# Influences of Beta-Alanine and l-Histidine Supplementation on Growth Performance, Meat Quality, Carnosine Content, and mRNA Expression of Carnosine-Related Enzymes in Broilers

**DOI:** 10.3390/ani11082265

**Published:** 2021-07-31

**Authors:** Bo Qi, Jing Wang, Meng Hu, Youbiao Ma, Shugeng Wu, Guanghai Qi, Kai Qiu, Haijun Zhang

**Affiliations:** 1Key Laboratory of Feed Biotechnology, Ministry of Agriculture and Rural Affairs, National Engineering Research Center of Biological Feed, Feed Research Institute, Chinese Academy of Agricultural Sciences, Beijing 100081, China; bo.qi@live.vu.edu.au (B.Q.); wangjing@caas.cn (J.W.); 18511602275@163.com (M.H.); myb0514@126.com (Y.M.); wushugeng@caas.cn (S.W.); qiguanghai@caas.cn (G.Q.); qiukai@caas.cn (K.Q.); 2College of Health and Biomedicine, Victoria University, Footscray, Victoria 3011, Australia

**Keywords:** beta-alanine, broiler, dipeptide, l-histidine, mRNA expression

## Abstract

**Simple Summary:**

In recent years, much attention has been paid to developing functional meat, which contains more functional peptides to impart health benefits. Poultry meat is a good source of imidazole dipeptides (carnosine and its derivative anserine), which are active endogenous constituents and may convey versatile physiological functions to promote health conditions. Carnosine is synthesized from l-histidine and beta-alanine. Dietary addition of histidine and/or beta-alanine may elevate the carnosine content in broiler meat. The current study further investigated the interaction of l-histidine and beta-alanine supplementation on carnosine content, meat quality, and gene expression of carnosine-related enzymes in broilers, which can facilitate a better understanding of the relationship between l-histidine and beta-alanine in carnosine synthesis.

**Abstract:**

The current study investigated the effect of dietary l-histidine and beta-alanine supplementation on growth performance, meat quality, carnosine content, and gene expression of carnosine-related enzymes in broilers. A two-factor design was adopted in this study. A total of 640 1-day-old male broilers were assigned to eight treatments with factorial arrangement containing four levels of l-histidine (0, 650, 1300, or 1950 mg/kg) and two levels of beta-alanine (0 or 1200 mg/kg) supplementation; 0 mg/kg histidine and/or 0 mg/kg were treated as control groups. Each treatment including eight replicates with 10 birds each and the feeding trial lasted for 42 days. Dietary supplementation with l-histidine and beta-alanine did not affect average daily gain (ADG), average daily feed intake (ADFI), and feed conversion ratio (FCR) of broilers during the grower (22–42 days) and the entire phase (1–42 days), compared with the control group (*p* > 0.05). The only exception was a significantly reduced ADG in the 1950 mg/kg l-histidine group in the starter period (1–21 days, *p* < 0.05). l-Histidine at 1950 mg/kg significantly decreased redness (a*) and yellowness (b*) values of the meat at 45 min postmortem (*p* < 0.05), whereas it increased b* value and pH in breast muscle at 24 h postmortem. Moreover, dietary supplementation with beta-alanine alone or combination with l-histidine significantly increased ΔpH in breast muscle (*p* < 0.01). Dietary l-histidine markedly increased total superoxide dismutase activity and total antioxidant capacity (T-AOC) both in breast muscle (*p* < 0.01) and in plasma (*p* < 0.01), and it decreased malondialdehyde (MDA) concentration in breast muscle (*p* < 0.01). Dietary addition of beta-alanine, alone or combination, significantly increased T-AOC in breast muscle (*p* < 0.01) and markedly decreased MDA content both in breast muscle and in plasma (*p* < 0.01). Addition of l-histidine and beta-alanine significantly increased muscle peptide (carnosine and anserine) content (*p* < 0.05) and upregulated the expression of carnosine synthase, transporter of carnosine/ l-histidine, and l-histidine decarboxylase genes (*p* < 0.05), with greater change occurring in the combination group of 1300 mg/kg l-histidine and 1200 mg/kg beta-alanine. Overall, dietary l-histidine and beta-alanine could improve meat quality and antioxidant capacity, enhance the carnosine and anserine content, and upregulate the gene expression of carnosine synthesis-related enzymes in broilers.

## 1. Introduction

Concurrent with increased social development and improved living standards, the improvement of diet composition has become a key factor to improve the health status and welfare of animals, as well as to enhance productivity in livestock [1] and performance in athletic species [2]. Today, there is a worldwide attempt to reduce antibiotic use in poultry production, which has caused increased microbial resistance to antibiotics and residues in animal products that can be harmful to consumers. Carnosine and anserine have antioxidation and antiaging properties, including better maintenance of muscle strength and pH buffering properties, which play an important role in stability, fatigue resistance, etc. [3]. Therefore, the development of improved and functional meat products (e.g., with elevated carnosine) may provide lifelong benefits and improve global health outcomes in aging populations. Carnosine is predominantly found in skeletal muscles and brain tissues [4,5] and is synthesized from l-histidine and beta-alanine by carnosine synthetase [6]. The concentration of carnosine is influenced by different factors such as animal species, sex, age, muscle fiber type, dietary composition and feeding methods (watering or feeding directly), and management [7,8,9,10,11].

Supplementation with 1 g l-histidine/kg of feed could elevates carnosine and anserine in chicken breast muscle by 64% and 10%, respectively [11]. Dietary addition of 22 mmol/kg (about 1.960 g/kg) beta-alanine increased carnosine content by 67% in chicken meat [12]. Moreover, dietary supplementation with histidine alone or with beta-alanine may increase carnosine concentrations of breast muscle [13]. A study in our laboratory showed that a supplemented diet of 1196 mg/kg beta-alanine produced the highest carnosine content in breast muscle in broilers, significantly improved the growth performance, and increased the gene expression of the carnosine synthetase and taurine transporter (a beta-alanine transporter) in breast muscle in broilers [14]. A previous study found similar results in mice with beta-alanine supplementation, which increased the expression of carnosine synthetase and taurine transporter gene [6]. However, knowledge on the effect of the interaction of histidine alone or combination with beta-alanine on the gene expression of carnosine-related enzymes is limited in broilers. Therefore, the main purpose of the current study was to explore the interaction of dietary fortification with l-histidine and beta-alanine on growth performance, meat quality, carnosine level, and gene expression of carnosine-related enzymes in broilers.

## 2. Materials and Methods

All experimental procedures were reviewed and approved by the Animal Care and Use Committee of the Feed Research Institute of the Chinese Academy of Agricultural Sciences (FRI-CAAS20181112).

### 2.1. Diets and Design of Experiment

A total of 640 1-day-old male Arbor Acre broilers were assigned to eight treatment groups with factorial arrangement containing four levels of l-histidine supplementation (0, 650, 1300, or 1950 mg/kg) and two levels of beta-alanine (0 or 1200 mg/kg). Each treatment consisted of eight replicates with 10 birds. All birds were raised in a battery cage (length × depth × height: 1.3 m × 0.7 m × 0.5 m) in an environmentally controlled room with continuous incandescent white light throughout the experiment. The room temperature was set to 33 °C for the first 3 days, and then the temperature dropped by 2 °C each successive week until it settled at 24 °C. Chicks had free access to feed and fresh water during the entire feeding trial. The composition of the experimental diet and the nutrient levels are presented in Table 1. All chickens were raised in accordance with the regulations of the Arbor Acre Broiler Commercial Management Guide [15].

### 2.2. Growth Performance

Body weight (BW) and feed intake were recorded during the starter phase (1 to 21 days), the grower phase (22 to 42 days), and the entire phase (1 to 42 days). ADG, ADFI, and FCR were calculated. On days 21 and 42, two birds at average BW from each replicate group were fasted for 12 h, weighed, and exsanguinated. Blood samples were collected (between 8:00 and 9:00 a.m. on trial day) in heparinized centrifuge tubes during bleeding from the left jugular vein and then immediately centrifuged at 1800× *g* for 15 min at 4 °C. The plasma samples were stored at −20 °C until analysis. The right side of breast muscle was subsequently removed to determine meat color and muscle pH. All samples were stored at 4 °C until analysis.

### 2.3. Meat Quality

#### 2.3.1. pH and Meat Color

Muscle pH values at 45 min (pH_45 min_) and 24 h (pH_24 h_) postmortem were determined using a calibrated electronic pH meter (CyberScan pH 310, Eutech Instruments Pte. Ltd., Singapore). Each sample was measured in triplicate at different positions within the muscle, and the average value was calculated as the result. The value of pH decline within 24 h postmortem (ΔpH) was calculated as ΔpH = pH_24 h_ − pH_45 min_. At 24 h after slaughter, meat color was measured in duplicate using a Chroma Meter (Chroma Meter WSC-S, Shanghai Precision and Scientific Instrument Co., Shanghai, China). Color was reported according to the CIE-Lab trichromatic system as lightness (L*), redness (a*), and yellowness (b*) values [16].

#### 2.3.2. Drip Loss

Right pectoralis muscles, which were trimmed into regular-shaped fillets (30 ± 1.5 g, wet weight, W_1_), were placed in a zip-sealed plastic bag and then filled with liquid nitrogen to avoid oxidation, evaporation, and mutual extrusion. All bags were stored at 4 °C for 24 h and reweighed (W_2_). Drip loss was calculated as (%) = (W_1_ − W_2_)/W_1_ × 100% (ZHANG 2009).

#### 2.3.3. Cooking Loss

After the drip loss, the samples remained in storage at 4 °C until 72 h postmortem. At 72 h postmortem, samples were removed, the water was wiped off the surface, and samples were weighed once more (W_3_). Next, the samples were placed in zip-sealed polyethylene bags and heated in a water bath at 85 °C for 20 min, cooled under running water to ambient temperature, dried, and weighed (W_4_). Cooking loss was calculated as (%) = W_3_ − W_4_/W_3_ × 100% (XU 2011).

#### 2.3.4. Shear Force

After measuring the cooking loss, the samples were stored at 4 °C until 96 h postmortem and then used for measuring the shear force value. Each sample was trimmed into three stripes (length × depth × height: 2 cm × 1 cm × 1 cm), each sample was cut three times at different locations, and the final results were expressed as the average value of nine cuts.

### 2.4. Antioxidant Indices in Breast Muscle and Plasma

About 0.5 g of frozen muscle was cut and homogenized for 2 min with 4.5 mL of 0.9% iced saline and then centrifuged at 3500× *g* for 10 min at 4 °C. The supernatant was collected and divided into small tubes and stored at −20 °C for analyzing antioxidant parameters, which included total superoxide dismutase (T-SOD) activity, total antioxidant capacity (T-AOC), and malondialdehyde (MDA) content. All parameters mentioned above were analyzed in muscle and plasma using standard commercial kits (Nanjing Jiancheng Bioengineering Institute, Nanjing, China) and a molecular microplate reader (Molecular Device LLC., San Jose, CA, USA). Each sample was measured in triplicate.

### 2.5. Dipeptide Content in Breast and Gene Expression of Carnosine Synthesis-Related Enzymes

#### 2.5.1. Dipeptide Content

The contents of carnosine and anserine were determined using an A300 automatic amino-acid analyzer (Membra Pure, Bodenheim, Germany). Samples were combined with 200 μL of sulfosalicylic acid (10%) to precipitate proteins kept at 4 °C for 1 h, and then centrifuged at 14,500× *g* for 15 min. The supernatant was filtered via a 0.45 μm pore size filter membrane after proper dilution (Millipore Co., Bedford, MA) and then directly injected into the analyzer for free amino-acid measurement. Next, 20 μL of ninhydrin, used as a trimethylbenzene substrate, was injected into the analyzer at 0.09 mL/min flow rate. Detection was performed by UV absorbance at wavelengths of 570/440 nm used for detection.

#### 2.5.2. Total RNA Extraction and cDNA Synthesis

Breast muscle tissues were homogenized with Trizol Reagent (Invitrogen, Carlsbad, CA, USA), and total RNA was extracted conforming to the manufacturer’s manual. The concentrations of RNA samples were quantified using the NanoDrop ND 1000 spectrophotometer (Thermo Fisher Scientific, Waltham, MA, USA). Total RNA (about 1 mg) extracted from each sample was utilized to synthesize the first-strand cDNA using the TIANGEN Quantscript RT kit, following the manufacturer’s instructions (TIANGEN Biotech Co. Ltd., Beijing, China). Procedures for RNA preparation conformed to the Minimum Information for Publication of Quantitative Real-Time (RT) PCR Experiments guidelines.

#### 2.5.3. Quantitative of Gene Expression

Gene expression of carnosine synthase (*CARNS*), carnosinase (*CNDP2*), carnosine/L-histidine transporters (*PHT1*), proton-coupled oligopeptide transporters (*PEPT1*, *PEPT2*), and l-histidine decarboxylase (*HDC*) was determined using a RT PCR kit according to the manufacturer’s instructions (Real MasterMix-SYBR Green; TIANGEN, China). Reactions of RT quantitative PCR were performed using an i-Cycler iQ5 multicolor RT PCR detection system (Bio-Rad, CA, USA), and the protocol used was as follows: 95 °C for 5 min; 40 cycles of 95 °C for 10 s, 60 °C for 30 s, and 72 °C for 30 s; final extension at 72 °C for 5 min.

The melting curve was recorded at 60 °C. The primers used are shown in Table 2. The amplification efficiency of each gene was validated by drawing a standard curve through four serial dilutions of cDNA. For analyses, relative quantification was applied with β-actin considered as the internal control. A bird sample chosen from control group was used as the internal sample. Protocols were done in triplicate. The relative mRNA expression levels of *CARNS*, *CNDP2*, *PHT1*, *PEPT1*, *PEPT2*, and *HDC* were calculated using the ∆∆C_t_ method, followed by Primers for RT PCR analysis [17,18].

### 2.6. Statistical Analysis

The normality of the data and the homogeneity of variances were tested in SPSS 16.0 for Windows (SPSS Inc., Chicago, IL, USA); the interactions between l-histidine and beta-alanine were analyzed with covariance analysis using the generalized linear models (GLM) of SPSS software. Then, the statistical significance of comparisons between the means of l-histidine and beta-alanine was further assessed using covariance analysis, as the interaction between the l-histidine and beta-alanine was not significant. The treatment effects were considered significant at *p* < 0.05, whereas a trend for a treatment effect was noted at 0.05 < *p* < 0.10.

## 3. Results

### 3.1. Growth Performance

The effect of dietary l-histidine and beta-alanine on growth performance is listed in Table 3. BW, ADFI, ADG, and FCR were not influenced by dietary beta-alanine supplementation throughout the entire period (*p* > 0.05). Dietary l-histidine addition markedly decreased ADG (*p* < 0.01), whereas it tended to reduce BW (0.05 < *p* < 0.1) and improve FCR (0.05 < *p* < 0.1) during the starter period (days 1 to 21). No notable differences in growth performance were found among all l-histidine treatments during the growth period (days 22 to 42, *p* > 0.05) and the entire period (days 1 to 42, *p* > 0.05). However, no significant differences were observed in response to the interaction of l-histidine and beta-alanine throughout the trial (*p* > 0.05).

### 3.2. Meat Quality

The effect of dietary l-histidine and beta-alanine on meat quality is shown in Table 4. At 45 min postmortem, dietary l-histidine supplementation at 1300 mg/kg or 1950 mg/kg significantly decreased a* and b* values (*p* < 0.05). Moreover, dietary supplementation of 1950 mg/kg l-histidine increased the pH_45 min_ and b*_24 h_ value of breast muscle (*p* < 0.05). However, the L* value at 45 min or 24 h did not vary in response to dietary l-histidine addition (*p* > 0.05). Furthermore, dietary beta-alanine supplementation significantly increased the ΔpH value (*p* < 0.05). Furthermore, the interaction of l-histidine and beta-alanine significantly decreased the L*_45 min_ value and increased the ΔpH_24 h_ and a*_24 h_ values of breast muscle in the current study (*p* < 0.05). However, no notable differences were observed in shear force, drip loss, and cooking loss in response to either l-histidine or beta-alanine supplementation in the diet (*p* > 0.05).

### 3.3. Antioxidant Indices in Breast Muscle and Plasma

The effects of dietary l-histidine and beta-alanine on antioxidant parameters (T-AOC, MDA, and T-SOD) in breast meat and plasma are illustrated in Table 5. Compared to the control, T-AOC and T-SOD activity in all l-histidine treatments and the interaction of l-histidine and beta-alanine groups was significantly increased both in breast meat and in plasma at day 42 (*p* < 0.05). Dietary beta-alanine supplementation elevated T-AOC in breast meat compared with the control group (*p* < 0.05). MDA concentrations were markedly decreased in all supplemental groups in breast meat (*p* < 0.05) and only in the beta-alanine treatment groups in plasma (*p* < 0.05).

### 3.4. Dipeptide Content in Breast and Gene Expression of Carnosine Synthesis-Related Enzymes

The effects of dietary l-histidine and beta-alanine on carnosine, anserine, and dipeptide are indicated in Figure 1. The contents of carnosine and anserine were significantly increased in all l-histidine-containing treatments at day 42 (*p* < 0.05), especially in the 1300 mg/kg l-histidine + 1200 mg/kg beta-alanine group. Moreover, dietary beta-alanine significantly increased the carnosine concentration (*p* < 0.05) and tended to increase the anserine levels (0.05 < *p* < 0.1). However, no significant difference was observed in dipeptide in response to beta-alanine supplementation in the diets (*p* > 0.05).

Table 6 shows the mRNA expression of *HDC*, *PHT1*, *PEPT1*, *PEPT2*, *CNDP1*, and *CARNS* in breast muscle. Dietary l-histidine supplementation significantly increased the expression of *HDC, PEPT1*, and *CARNS* (*p* < 0.05) and tended to improve the mRNA expression of *PHT1* in breast muscle (0.05 < *p* < 0.1). Furthermore, the mRNA expression of *CARNS* was improved in response to beta-alanine supplementation (*p* < 0.05). However, only the expression of *PHT1* was improved by the interaction of l-histidine and beta-alanine in breast muscle (*p* < 0.05).

## 4. Discussion

Dietary l-histidine supplementation decreased the growth performance throughout the entire experimental period. These results are consistent with a previous study [19]. In the current study, chickens fed diets containing l-histidine from 650 to 1950 mg/kg (0.065% to 0.195%) significantly reduced the ADG, improved the FCR, and tended to decrease the body weight during the starter growth period. Supplementation with 0.3% l-histidine in the diet reduced the body weight, ADG, and ADFI by 13.6%, 13.6%, and 10.5%, respectively, during the period of 10 to 28 days [20]. However, there was no effect on body weight and FCR in response to 0.18% l-histidine supplementation at day 42 in turkey [21]. At the same time, there was no significant influence in response to beta-alanine supplementation. Some similar results were obtained from studies with the addition of beta-alanine in chickens [13], mice [6], and pigs [22].

In the current study, diets supplemented with l-histidine reduced feed efficiency by decreasing ADG. A possible reason is that l-histidine is purported to be toxic in high supplementation dosages (2000 mg/kg BW per day) and has demonstrated retarded growth rates and hypercholesterolemia [23,24]. In addition, l-histidine could influence body weight through its conversion into neuronal histamine, which could suppress the food intake or appetite [25]. This concept is supported by an earlier study, which revealed that chickens depress growth in response to an increase in histamine [26].

Meat quality is used to describe the properties and perception of meat, and good meat quality can improve consumers’ acceptance of meat products. The meat color is a consequence of physiological, biochemical, and microbiological changes in muscles and is an important parameter related to muscle pH [27]. In the present study, supplementation with 1950 mg/kg histidine significantly decreased the a^*^ value and b^*^ value at 45 min postmortem, yet it increased the pH value at 24 h postmortem. Meanwhile, beta-alanine alone or combined with l-histidine supplementation markedly increased the pH_45 min_ and ΔpH_24 h_ values at day 42. A previous study reported that the breast muscle had a lower degree of a^*^ value in response to histidine supplementation in diets [11], which is similar to our result. The possible reason may be related to changes in histamine content. Poultry meat with high pH has been associated with high water-binding capacity and subsequently improved meat quality [28]. Meat color varies because of the different concentrations of pigments; one of the principal pigments in poultry is hemoglobin [29], and the level of hemoglobin was shown to have a positive connection with histamine [30].

The values of shear force, drip loss, and cooking loss in breast muscle were not affected by dietary l-histidine or/and beta-alanine supplementation in broilers [11], which is similar to the current study. However, dietary supplementation with 200 or 400 mg/kg carnosine significantly decreased the shear force of breast in broilers because carnosine could inhibit the process of lipid peroxidation. The latter could cause Ca^2+^ pump uncoupling and Ca-ATPase inactivation, leading to an increase in Ca^2+^ levels and a reduction in shear force value [31]. Moreover, muscles at pH ≥ 6.0 are characterized by lower protein denaturation, whereas muscles at pH ≤ 6.0 undergo greater protein denaturation [30]. In the present study, the increased pH or ΔpH in all treatments implied that beta-alanine and/or l-histidine may slow down protein denaturation. Meanwhile, carnosine is a functional dipeptide found in high concentrations in skeletal muscle [3] and is considered a potent buffer due to the pKa of its imidazole ring (6.83) [32]. The concentration of carnosine was markedly increased in breast muscle in response to all treatments in the current study, which would help to improve meat quality.

In the present study, dietary l-histidine addition significantly elevated the T-AOC and SOD contents both in muscle and in plasma, whereas it decreased the MDA content in breast muscle, especially in the 1300 mg/kg l-histidine + 1200 mg/kg beta-alanine group. These results are similar to a previous study [33], which reported that dietary histidine increased the activity of SOD of broilers both in breast muscle and in plasma. l-Histidine demonstrates strong antioxidant characteristics and may react with free radicals and the carbonylation of proteins in the body to resist oxidation or the action of metal ions that promote the activity of oxidase [34].

Furthermore, dietary supplementation of carnosine increased the activity of T-AOC, yet it decreased the MDA content in breast muscle in broilers at day 42 [35]. It was reported that beta-alanine had a lower level of thiobarbituric acid reactive substances (mg MDA/kg of tissue) than the control group [36]. Moreover, the interaction of l-histidine and beta-alanine decreased MDA in breast muscle and increased T-AOC and T-SOD both in breast muscle and in plasma. The improved antioxidant abilities may result from the increased concentration of carnosine, which has a strong antioxidant capacity [31,37]. Therefore, the combination of 1300 mg/kg l-histidine and 1200 mg/kg beta-alanine led to superior muscle antioxidant ability in the current study.

Carnosine is synthesized from two precursors (l-histidine and beta-alanine) via carnosine synthase [38]. Anserine is a methylated form of carnosine, which was found at high levels in the muscles of salmon and tuna, which are migratory fish [39]. In the present study, supplementation with 1300 mg/kg (0.13%) l-histidine significantly enhanced the concentration of carnosine and anserine (31.6% and 6.75%, respectively*)*, which is similar to previous studies showing that the content of carnosine and anserine increased (64% and 10%, respectively) in response to 0.1% l-histidine supplementation in breast muscle in broilers [20] and that dietary supplementation with 0.18% l-histidine increased carnosine and anserine by 35.24% and 23.33%, respectively [32]. Moreover, dietary beta-alanine or combinative supplementation with l-histidine markedly increased carnosine content in breast muscle; these results corroborate a previous finding that 1.2% beta-alanine or 1.8% carnosine supplementation significantly increased carnosine concentration in soleus in mice [6]. Several studies revealed that dietary beta-alanine supplementation could influence the concentration of histidine [40,41], which may lead to imbalance in the process of carnosine production and influence the carnosine levels; however, beta-alanine and histidine were not detected in breast muscle in the current results, which necessitates further study.

Furthermore, the gene expression of the enzymes and transporters related to carnosine synthesis were explored in skeletal muscle (*CARNS*, *CNDP2*, *PEPT1*, *PEPT2*, *PHT1*, and *HDC*); only the mRNA expression of *CARNS*, *PEPT1*, and *HDC* was significantly upregulated in response to l-histidine and beta-alanine supplementation in the current study. It is expected that the synthesis of carnosine would be limited by the expression of *CARNS* [42], whereas the upregulation of *CARNS* gene could promote the accumulation of carnosine in breast muscle in broilers [6]. *PEPT1* can transport both l-histidine and carnosine in skeletal muscle [6]; hence, competition for transporter *PEPT1* may exist between histidine and carnosine. Dietary addition of l-histidine stimulates the expression of *HDC*, which is an enzyme responsible for the decarboxylation of histidine to produce histamine [43]. Histamine may potentially negatively affect body weight (as mentioned earlier). However, the elevation of *HDC* expression implies that the possibility of carnosine degradation to histidine and subsequent decarboxylation inside the skeletal muscle cell may occur. Therefore, carnosine levels in skeletal muscle increased through complicated interactions among *CARNS*, *PEPT1*, and *HDC*.

## 5. Conclusions

Dietary addition of l-histidine and beta-alanine could improve meat quality by increasing T-AOC and carnosine content and decreasing MDA level in breast muscle. The increase in muscular carnosine content was likely due to the upregulation of mRNA expression of carnosine synthesis-related enzymes and transporters (*CARNS*, *PEPT1*, and *HDC*) in broilers following 42 days of dietary l-histidine and beta-alanine supplementation. Moreover, the interaction of 1300 mg/kg l-histidine and 1200 mg/kg beta-alanine improved the meat quality, antioxidant capacity, and carnosine levels in the breast muscle of broilers with no adverse effects on growth performance.

## Figures and Tables

**Figure 1 animals-11-02265-f001:**
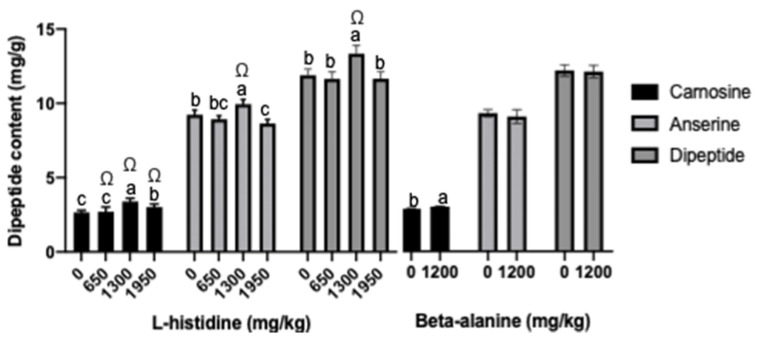
Effects of dietary l-histidine and beta-alanine on content of carnosine and anserine in broiler chicks at day 42 ^1^. ^1^ *n* = 8 replicates per treatment. ^a–c^ Means in a column lacking a common superscript differ significantly (*p* < 0.05). ^Ω^ Means in a column lacking a common superscript differ significantly (interaction of l-histidine and beta-alanine) (*p* < 0.05).

**Table 1 animals-11-02265-t001:** Composition of the experimental diet and the nutrient levels (air-dried basis, %).

Item	Starter Period (1–21 Days)	Grower Period (22–42 Days)
Components, %		
Corn	59.22	62.20
Soybean meal (47%)	34.37	30.84
Vegetable oil	2.27	3.26
Dicalcium phosphate	1.82	1.55
Limestone	1.31	1.23
Salt	0.30	0.30
dl-Methionine (98%)	0.24	0.19
l-Lysine-HCl (78%)	0.09	0.07
l-Threonine (98)	0.06	0.04
Vitamin premix ^1^	0.02	0.02
Mineral premix ^2^	0.20	0.20
Choline chloride (50 %)	0.10	0.10
Total	100	100
Calculated nutrient levels		
AME, MJ/kg	12.35	12.77
Crude protein, %	21.5	20.00
Calcium, %	1.00	0.90
Total phosphorus, %	0.69	0.63
Available phosphorus, %	0.45	0.40
Lysine, %	1.21	1.10
Methionine, %	0.55	0.48
Methionine + cysteine, %	0.88	0.80
Threonine, %	0.86	0.78

^1^ The vitamin premix supplied the following per kg of complete feed: vitamin A, 12,500 IU; vitamin D_3_, 2500 IU; vitamin K_3_, 2.65 mg; vitamin B_1_, 2 mg; vitamin B_2_, 6 mg; vitamin B_12_, 0.025 mg; vitamin E, 30 IU. ^2^ The mineral premix supplied the following per kg of complete feed: Cu, 8 mg; Zn, 75 mg; Fe, 80 mg; Mn, 100 mg; Se, 0.15 mg; I, 0.35 mg; biotin, 0.0325 mg; folic acid, 1.25 mg; pantothenic acid, 12 mg; niacin, 50 mg.

**Table 2 animals-11-02265-t002:** Primers used in qPCR analysis in broiler chicks.

Genes	Forward Primer (5′–3′)	Reserve Primer (3′–5′)	T_m_ (°C)	Gene ID
*CARNS*	CTGGAGGGGTCAGCAAGAG	CTGTCGTAGGGCAGGAAGGT	62	100359387
*CNDP2*	CACCTCACCTTCTGGCTTGT	ACATGCTTCCCTCTTCTCCA	62	421013
*PEPT1*	TGTCACTGGGCTGGAGTTTT	AGCAAGGCAGCAAAGAGAAC	60	378789
*PEPT2*	GTGGGGTTCAGACATGGAAG	GGCCAGACCTGTAATGGAGA	62	424244
*PHT1*	CTGGCAGAGGACAAACACAA	ACTCGCTGCACTCAATTTCC	60	416808
*HDC*	GGCAGGCTCTTCCTTATTCC	GCAGTGCGTTGAATGATGTT	62	425454
*β-actin*	TGACAATGGCTCCGGTATGT	TCTTTCTGGCCCATACCAAC	60	396526

Note: *CARNS*, carnosine synthase; *CNDP2*, carnosinase-2; *PEPT1/PEPT2*, proton-coupled oligopeptide transporters; *PHT1* carnosine/histidine transporters; *HDC*, histidine decarboxylase.

**Table 3 animals-11-02265-t003:** Effect of dietary l-histidine and beta-alanine supplementation on growth performance in broiler chicks ^1^.

L-histidine (mg/kg)		Beta-alanine (mg/kg)		BW	ADFI	ADG	FCR	BW	ADFI	ADG	FCR	ADFI	ADG	FCR
				1–21 Days				22–42 Days				1–42 Days		
0		0		819.66	53.52	37.71	1.42	2411.86	146.26	75.87	1.94	90.32	52.81	1.71
0		1200		848.57	53.23	39.07	1.40	2562.31	150.97	79.18	1.91	92.59	52.99	1.68
650		0		843.37	54.61	38.88	1.41	2416.83	148.21	73.27	2.04	92.87	52.92	1.76
650		1200		825.00	53.91	37.86	1.45	2510.90	147.24	78.73	1.87	92.08	54.13	1.70
1300		0		809.61	54.27	36.93	1.47	2429.88	143.93	73.58	1.96	90.88	51.77	1.75
1300		1200		801.53	53.51	35.99	1.46	2391.71	141.65	71.75	1.98	87.59	50.13	1.75
1950		0		815.89	53.28	37.51	1.42	2553.80	146.97	78.61	1.88	91.98	54.41	1.69
1950		1200		786.40	52.26	34.69	1.52	2438.17	138.10	73.94	1.89	86.29	50.14	1.72
Pooled SEM				5.860	0.475	0.332	0.010	25.497	1.353	1.076	0.017	0.858	0.523	0.009
**Source**				***p*-value**										
l-histidine				0.085	0.708	0.024	0.076	0.642	0.252	0.432	0.279	0.430	0.353	0.119
Beta-alanine				0.567	0.469	0.204	0.129	0.658	0.497	0.794	0.214	0.279	0.285	0.355
l-histidine × beta-alanine				0.330	0.995	0.185	0.135	0.247	0.373	0.332	0.197	0.404	0.277	0.365
Main Effect	l-histidine		0	834.11	53.38	38.39 ^a^	1.41	2487.09	148.62	77.52	1.93	91.46	52.9	1.69
			650	834.19	54.26	38.37 ^a^	1.43	2463.87	147.73	76.00	1.96	92.48	53.52	1.73
			1300	805.57	53.89	36.46 ^ab^	1.47	2410.79	142.79	72.67	1.97	89.24	50.95	1.75
			1950	801.15	52.77	36.10 ^b^	1.47	2495.99	142.54	76.28	1.89	89.14	52.27	1.71
	Beta-alanine		0	822.13	53.92	37.76	1.43	2453.09	146.34	75.33	1.96	91.51	52.97	1.73
			1200	815.38	53.23	36.90	1.46	2475.77	144.49	75.90	1.91	89.64	51.85	1.71

^1^ *n* = 8 replicates per treatment. ADG (g/day) = average daily gain; ADFI (g/day) = average daily feed intake; FCR (feed/gain = g:g) = feed conversion ratio; BW (g) = body weight. ^a,b^ Means in a column lacking a common superscript letter differ significantly (*p* < 0.05).

**Table 4 animals-11-02265-t004:** Effects of dietary l-histidine and beta-alanine on meat quality of broiler chicks at day 42 ^1^.

l-histidine (mg/kg)		Beta-alanine (mg/kg)		L	a	b	pH	L	a	b	pH	△pH	Shear Force	Drip Loss	Cooking Loss
				45 min				24 h							
0		0		49.49	8.35	16.11	6.13	56.97	8.21	11.58	5.72	0.41	17.45	5.75	8.04
0		1200		52.00	9.85	16.10	6.29	58.72	7.89	12.23	5.77	0.59	16.87	5.60	8.17
650		0		53.58	8.12	14.18	6.19	58.64	7.31	11.88	5.89	0.30	19.17	5.21	8.59
650		1200		51.20	7.73	15.47	6.35	58.60	7.98	11.98	5.77	0.58	18.27	7.23	8.51
1300		0		49.85	6.02	14.33	6.36	56.54	7.76	11.94	5.85	0.50	20.58	6.88	8.68
1300		1200		49.48	6.41	11.61	6.27	55.97	8.18	12.04	5.83	0.43	20.57	5.79	8.29
1950		0		49.55	6.13	14.03	6.31	56.00	8.15	13.23	5.89	0.48	21.46	5.52	8.52
1950		1200		50.56	5.60	13.49	6.32	58.26	8.03	13.46	5.84	0.48	18.29	3.80	7.91
Pooled SEM				0.244	0.013	0.302	0.016	0.317	0.052	0.160	0.012	0.018	0.644	0.335	0.067
**Source**				***p*-value**											
l-histidine				0.001	<0.001	0.004	0.095	0.065	0.016	0.005	0.005	0.760	0.348	0.066	0.602
Beta-alanine				0.691	0.433	0.419	0.058	0.186	0.127	0.400	0.144	0.010	0.412	0.633	0.374
l-histidine × beta-alanine				0.007	0.086	0.138	0.021	0.331	0.004	0.917	0.089	0.005	0.872	0.054	0.768
Main effect	l-histidine		0	50.74 ^ab^	9.10 ^a^	16.10 ^a^	6.21	57.85	8.05 ^a^	11.90 ^b^	5.75 ^b^	0.50	17.09	5.67	8.11
			650	52.39 ^a^	7.92 ^b^	14.83 ^ab^	6.27	58.62	7.65 ^b^	11.93 ^b^	5.83 ^ab^	0.44	18.72	6.08	8.54
			1300	49.67 ^b^	6.22 ^c^	12.97 ^b^	6.31	56.25	7.97 ^ab^	11.99 ^b^	5.84 ^a^	0.47	20.58	6.37	8.51
			1950	50.06 ^b^	5.87 ^c^	13.76 ^b^	6.32	57.13	8.09 ^a^	13.35 ^a^	5.86 ^a^	0.48	19.35	4.66	8.25
	Beta- alanine		0	50.62	7.16	14.66	6.25	57.04	7.86	12.16	5.84	0.42 **	19.55	5.84	8.52
			1200	50.81	7.40	14.17	6.31	57.89	8.02	12.43	5.80	0.52 *	18.19	5.49	8.24

^1^ *n* = 8 replicates per treatment. ^a–c^ Means in a column lacking a common superscript letter differ significantly (*p* < 0.05). *, ** Means in a column lacking a common superscript differ significantly (*p* < 0.05).

**Table 5 animals-11-02265-t005:** Effect of dietary l-histidine and beta-alanine on antioxidant parameters in breast muscle and plasma of broiler chicks at day 42 ^1^.

l-histidine (mg/kg)		Beta-alanine (mg/kg)		Breast			Plasma		
				T-AOC	MDA	T-SOD	T-AOC	MDA	T-SOD
0		0		0.22	0.92	45.94	3.55	2.26	149.48
0		1200		0.30	0.70	49.48	3.92	2.12	169.86
650		0		0.26	0.75	48.71	3.62	2.22	177.87
650		1200		0.26	0.76	45.31	4.93	2.10	167.61
1300		0		0.28	0.72	49.17	4.99	2.28	177.30
1300		1200		0.32	0.69	47.07	6.20	2.15	186.17
1950		0		0.29	0.60	51.37	5.63	2.28	206.40
1950		1200		0.26	0.64	51.26	4.67	2.23	158.79
Pooled SEM				0.005	0.008	0.335	0.127	0.017	2.343
**Source**				***p*** **-value**					
l-histidine				0.010	<0.001	<0.001	<0.001	0.253	0.004
Beta-alanine				0.024	0.003	0.444	0.063	0.002	0.133
l-histidine × beta-alanine				0.001	0.003	0.004	0.009	0.708	<0.001
Main effect	l-histidine		0	0.26 ^b^	0.81 ^a^	47.71 ^b^	3.74 ^c^	2.19	159.67 ^b^
			650	0.26 ^b^	0.75 ^b^	47.01 ^b^	4.28 ^bc^	2.16	172.74 ^a^
			1300	0.30 ^a^	0.71 ^b^	48.12 ^b^	5.59 ^a^	2.22	181.74 ^a^
			1950	0.28 ^ab^	0.62 ^c^	51.31 ^a^	5.15 ^ab^	2.25	182.59 ^a^
	Beta-alanine		0	0.26 **	0.75 *	48.80	4.45	2.26 *	177.76
			1200	0.29 *	0.70 **	48.28	4.93	2.15 **	170.61

^1^ *n* = 8 replicates per treatment. T-SOD (U/mL) = total superoxide dismutase; T-AOC (U/mL) = total antioxidant capacity; MDA (nmol/mL) = malondialdehyde. ^a–c^ Means in a column lacking a common superscript differ significantly (*p* < 0.05). *, ** Means in a column lacking a common superscript differ significantly (*p* < 0.05).

**Table 6 animals-11-02265-t006:** Effects of dietary l-histidine and beta-alanine on mRNA expression of carnosine synthesis-related enzymes in broiler chicks ^1^.

l-histidine (mg/kg)		Beta-alanine (mg/kg)		HDC	PHT1	PEPT1	PEPT2	CNDP2	CARNS
0		0		1.00	1.00	1.00	1.00	1.00	1.00
0		1200		0.94	1.41	1.09	1.41	1.02	1.07
650		0		1.06	1.62	1.58	1.13	1.44	1.00
650		1200		1.42	1.69	1.57	1.30	1.87	1.60
1300		0		2.24	1.92	1.60	1.30	1.51	1.57
1300		1200		1.61	1.48	1.51	1.48	2.30	1.95
1950		0		1.32	1.21	1.42	1.34	1.64	1.34
1950		1200		1.52	1.87	1.50	1.38	1.55	1.78
Pooled SEM				0.088	0.068	0.063	0.076	0.127	0.127
**Source**				***p*** **-value**					
l-histidine				0.004	0.060	0.017	0.743	0.100	<0.001
Beta-alanine				0.853	0.201	0.897	0.201	0.268	0.002
l-histidine × beta-alanine				0.232	0.038	0.951	0.863	0.609	0.397
Main effect	l-histidine		0	0.97 ^b^	1.21	1.04 ^b^	1.20	1.01	1.04 ^c^
			650	1.24 ^b^	1.65	1.57 ^a^	1.21	1.66	1.30 ^bc^
			1300	1.93 ^a^	1.70	1.55 ^a^	1.39	1.91	1.76 ^a^
			1950	1.42 ^ab^	1.54	1.46 ^ab^	1.36	1.60	1.56 ^ab^
	Beta-alanine		0	1.41	1.44	1.40	1.19	1.40	1.23 **
			1200	1.37	1.61	1.42	1.39	1.69	1.60 *

^1^ *n* = 8 replicates per treatment. *CARNS*, carnosine synthase; *CNDP2*, carnosinase-2; *PEPT1/PEPT2*, proton-coupled oligopeptide transporters; *PHT1* carnosine/histidine transporters; *HDC*, histidine decarboxylase. ^a–c^ Means in a column lacking a common superscript differ significantly (*p* < 0.05). *, ** Means in a column lacking a common superscript differ significantly (*p* < 0.05).

## Data Availability

None of the data were deposited in an official repository. This is an open access article. Anyone who needs to see raw data could email me (bo.qi@live.vu.edu.au).

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
