# Peer review of "Influences of Beta-Alanine and l-Histidine Supplementation on Growth Performance, Meat Quality, Carnosine Content, and mRNA Expression of Carnosine-Related Enzymes in Broilers"

_animals, 2021, doi:10.3390/ani11082265_

Round 1

Reviewer 1 Report

Authors satisfied the Reviewer concerns and modified the manuscript according to Reviewer suggestions and comments.

The manuscript was significantly improved.

Thus, the manuscript results suitable for publication.

Author Response

Thank you very much for your favorable comments.

Reviewer 2 Report

Dear Authors,

Thanks for the revised version. However, still there are a few issues that need to be addressed. 

  • Please add your explanation about functional meat into the text. Using only "functional meat" in the text could be confusing for readers. 
  • Regarding Meat quality parameters, the method and the result sections have been accordingly revised. However, I would recommend you to add a few lines in the discussion section even the data is insignificant. 

Best regards,  

Author Response

Dear editor & reviewers,

Again, thank you so much for your invaluable suggestions! Your suggestions gave us much to learn and helped improve our scientific writing to a great extent.

We had revised our manuscript with highlight green in the revised manuscript according to your suggestions. We also checked all references. We hope we could understand your questions correctly and have given the right answers in the revised manuscript. Thanks a lot.

Question 1: Please add your explanation about functional meat into the text. Using only "functional meat" in the text could be confusing for readers. 

Answer: Thank you very much for your kind advice, we revised and gave the explanation in Line 13-14. Please check it.

Question 2: Regarding Meat quality parameters, the method and the result sections have been accordingly revised. However, I would recommend you to add a few lines in the discussion section, even the data is insignificant. 

Answer: Dear reviewer, thank you very much for your considerate suggestion. We added more information about meat quality parameters, please check Line 306-312 in the discussion section and Line 446-447 in the references section. Thanks a lot!

This manuscript is a resubmission of an earlier submission. The following is a list of the peer review reports and author responses from that submission.

Round 1

Reviewer 1 Report

I think that the subject of the work is of interest and that the topic of the manuscript is appropriate for the Journal. The information is of significant interest to the Journal's readers. However, introduction section should be enriched in order to and to better justify the rationale of the study. Moreover, several missing information should be added in the methods section. The discussion section need to be improved in order to be more harmonic. In view of such consideration, I suggest that the study could be suitable for publication after major revision.

Specific Comments

The title as well as keywords accurately reflects the major findings of the work.

The abstract adequately summarize methodology, results, and significance of the study. However, Authors should indicate statistical analysis applied.

The introduction section falls within the topic of the study, however, Authors should enhance this section adding more information concerning the diet supplementation in veterinary field emphasizing the significant increase of interest showed by scientific community on diet improvement to enhance animal health status and welfare.  On this regards, at the beginning of introduction section before “Concurrent…”, Authors could write the following information and the related references “The improvement of diet composition become a key factor to improve the health status and welfare of animals as well as to enhance productivity in livestock (Monteverde V. et al., Journal of Applied Animal Research, 2017, 45: 615-618) and physical performance in athletic species (Piccione G. et al., Animal Production Science, 2019, 59 (2): 232-235). Nowadays there is a worldwide attempt to reduce antibiotic use in poultry production which cause increased microbial resistance to antibiotics and residues in animal products that can be harmful to consumers.”

The section of Materials and Methods is clear for the reader and it well describes the methods applied in the study.  However, Authors should check this section and correct many punctuation errors. In addition some important information should be added.

Have the enrolled animals been checked for parasitic infections?

Authors wrote: “The chicken house was maintained on a constant (23 h light: 1 h darkness) lightening program after the…” please add more information regarding the chicken house (e.g. the dimension) and the instrumentation used for the maintenance of lightening program as well as of temperature.

In the previous subsection, Authors wrote “The plasma samples were stored at –20°C until analysis.” My question is: Are the authors sure that the considered antioxidant indices can be analysed from plasma frozen at -20 °C and not at -80 °C?

Authors should indicate the instrument used for the assessment of the considered antioxidant parameters.

Moreover, Authors should indicate the time of blood sampling (hour of the day) from animals, since the parameters could follow a circadian rhythm; Authors should indicate the time between blood collection and centrifugation.

Regarding the assessment of gene expression, generally at least three reference genes should be included and used as potential internal controls for normalization of gene expression data. Why Authors used only one?

Did the Authors perform the analysis in duplicate?

Moreover, Authors wrote “The relative mRNA expression levels of CARNS, CNDP2, PHT1, PEPT1, PEPT2 and HDC were calculated using the “normalized relative quantification” method followed by Primers for RT PCR analysis.” Please, could you indicate a valid reference? Why a ΔΔ CT method was not applied?

Results section is clear and well written. The findings obtained in the study were well discussed and justified with appropriate references. However, I suggest to simplify the discussion section since it is too long and appear somewhat unclear for the reader. Probably the subdivision of the Discussion into subsections makes this section not very discursive. Therefore, I suggest to make the discussion section more harmonic  by deleting the subsection.

The conclusion section is clear, well written and Authors well summarized the results and the significance of the study. 

The tables are generally good and well represent the results of the study. However, if it is possible, I suggest to replace some tables in graphs to make the results more attractive and impactful.

Authors should check and standardize the references in the list according to journal guidelines.

Reviewer 2 Report

Dear Authors,

Please find my comments below:

Line 13. what is your mean functional meat? 

Line 17. Please remove the citation to Poultry Science, You should not cite your work here!

Simple summary. This section needs to be improved, the current shape does not look like a summary. Talk about your current study achievements, not other works. Best is to rewrite this section. 

Line 27. Better to provide a full name for the first time for example Average Daily Gain (ADG). Please apply to all

Line 91-93. Please remove the statement.  The experimental procedures were approved by the Animal Care and Use Committee of the Feed Research Institute of the Chinese Academy of Agricultural Sciences. This statement is already mentioned in lines 79-81. 

Line 104. 2 birds at average BW from each replicate group were fasted for 12 hours

I assume this was done randomly which the authors need to add a statement to explain this.

However, this could not be done randomly; the authors attempted to pick birds near the average BW. Investigator/s of the study could pick the better shape or healthier birds from a specific group. 

Table 4. Control missing? I believe there should be a 0 L-histi and 0 Beta. 

The manuscript is missing basic important data regarding the meat quality such as shear force, water holding capacity, or drip loss. I believe there is a lack of a strong connection between the meat quality results and antioxidants data.  Line 279-280 is an example. 

There is no statement in The meat quality and Antioxidants sections to connect the results together. As I mentioned earlier, missing basic important data could bring more questions about the results with no possible answers for them. 

The authors need to add more data regarding the meat quality to support the current results.

Best regards,